# Physico-Chemical Properties, Fatty Acids Profile, and Economic Properties of Raspberry (*Rubus idaeus* L.) Seed Oil, Extracted in Various Ways

**DOI:** 10.3390/plants12142706

**Published:** 2023-07-20

**Authors:** Audrone Ispiryan, Ramune Bobinaite, Dalia Urbonaviciene, Kristina Sermuksnyte-Alesiuniene, Pranas Viskelis, Astrida Miceikiene, Jonas Viskelis

**Affiliations:** 1Institute of Horticulture, Lithuanian Research Centre for Agriculture and Forestry, Kauno Str. 30, 54333 Babtai, Lithuania; dalia.urbonaviciene@lammc.lt (D.U.); pranas.viskelis@lammc.lt (P.V.); jonas.viskelis@lammc.lt (J.V.); 2Lithuanian Centre for Social Sciences, Institute of Economics and Rural Development, A. Vivulskio Str. 4A-13, 03220 Vilnius, Lithuania; kristina@agrifood.lt; 3Agriculture Academy, Vytautas Magnus University, Studentų Str. 11, 53361 Akademija, Lithuania; astrida.miceikiene@vdu.lt

**Keywords:** raspberry seed oil, sustainable products, extraction, fatty acids, by-products

## Abstract

In Europe, the green course is becoming increasingly relevant, and there are more and more suggestions for its improvement. The valorization of food waste attracts increasing attention and is one important current research area. The aim of this study was to examine oils from 16 raspberry variety seeds and to compare their yields and fatty acid contents. The next task was to extract oil from the raspberry variety ‘Polka’ by four different methods and to compare the yield, colors, fatty acids content and composition, and kinematic and dynamic viscosity. The last task was to analyze the economic profitability of oil extraction by different methods. This study demonstrates the potential of different varieties of raspberry by-products and shows the influence of different oil extraction methods on the fatty acid composition of the oil and the economic potential of such products. The analysis revealed that the predominating fatty acid in the raspberry variety ‘Polka’ seed oil was linoleic acid (44.0–44.8%), followed by α-linolenic acid (37.9–38.1%) and oleic acid (10.2–10.6%). Of the 16 raspberry cultivars tested, ‘Polka’ seed oil had the least linoleic (ω-6) (44.79%) and the most α-linolenic (ω-3) fatty acids and the best ratio of ω-6 to ω-3 fatty acids—1.2:1. Raspberry variety ‘Polka’ seed oil contains a lot of carotenoids; their total amount depending on the extraction method varies from 0.81 mg/100 g (extracted with subcritical CO_2_) to 3.25 mg/100 g (extracted with supercritical CO_2_). The oil yield can be increased by grinding the seeds into a finer fraction. The most expensive method of oil production is supercritical CO_2_ extraction, and the cheapest method with the fastest payback of equipment is the cold-pressing method. The results of the research have revealed the influence of different oil recovery methods on the yield of oil, the composition of the fatty acid, colors, and viscosity. The results are very important for producers wishing to commercialize raspberry seed oil.

## 1. Introduction

Due to various application possibilities and its economic impact, the valorization of food waste has attracted increasing attention. The valorization of food waste is one important current research area because it solves both environmental and economic problems. Fruit waste valorization has gained recent significance as it can be used as an important tool to meet sustainable development goals and help to combat the carbon footprint and greenhouse gas emissions that are mostly caused by these wastes. Although some fruit by-products contain even more bioactive ingredients than the original fruit itself, they are typically seen as waste and thrown away [1].

The red raspberry (*Rubus idaeus*), a species widely known for its edible fruits, is a fruit in the genus *Rubus*, grown as a perennial crop. There are about 200 raspberry species, and most of these have red berries (European) [2,3]. Raspberry is one of the oldest fruits, has medicinal purposes, and is the fourth most important berry crop in the world. From 2010 to 2020, raspberry production has increased from 373 thousand tons to 684 thousand tons in the world [4,5]. Raspberries are greatly used in food manufacturing for purees, juices, jams, wines, etc. Raspberry seeds are an important by-product in the production process of raspberry wines and juices, but usually discarded and thus underexploited. Various researchers have discovered that due to raspberry seeds’ high content of antioxidants, phenolic acids, flavonoids, polyphenols, and fibers, as well as the high amount of waste released during industry manufacturing, these by-products could be successfully recovered and used for different industry purposes. Furthermore, it is known by recent works that raspberries are particularly high not only in anthocyanin content, but also in total phenolic content [6,7,8].

During the processing, predominantly in the fruit and beverage industry for juice and jam production, a large number of their by-products (pomace, consisting mostly of the seeds) are produced. Since blackberry and raspberry seeds contain lipids, these by-products are very interesting as a raw material for oil manufacturing in small facilities. Therefore, berry oils are specialty oils and have been in demand on the market. These oils have a unique fatty acid profile, and they possess interesting minor components [9].

Raspberry seed components can be separated into two parts: oil and flour. The flour remains following oil extraction, which is performed by many methods, including solvent extraction and cold pressing. Seeds have very different and complex chemical compositions that are nutritionally grouped as macronutrients, micronutrients, and other components. Other components include other phytochemicals, such as phenolic antioxidants, which have demonstrated potential beneficial health properties. Seed oils also have other properties that include oxidative stability and color. These components and properties of seeds are valuable and need to be examined and reported, which may ultimately lead to increased crop values and increased farm gate profits for growers and processors [10].

The choice of a method to obtain maximum yield and highest purity varies according to the nature of the target compound. Numerous chemical and mechanical processes like solvent extraction, enzymes-assisted extraction, pulsed electric field-assisted extraction, and steam distillation are used for the extraction of compounds from plants [11,12,13]. In the last few years, supercritical fluid extraction (SFE) has received significant attention as a promising alternative to conventional technology for separation of various valuable compounds from natural sources [14,15,16,17]. The oil obtained in this way is much more expensive on the market than that obtained by cold pressing, and consumers have formed the opinion that such oil is correspondingly more valuable; its properties are exceptional [18].

Raspberries’ bioactive compounds are recommended for use not only in daily consumption but also for helping manage or prevent various human diseases such as cancer, diabetes, neurodegenerative disorders, and cardiovascular and heart disease [19,20,21,22]. The use of pomace in the food industry can create some opportunities to lower production costs and to create a new food source for human consumption [23,24]. 

This paper demonstrates the potential of raspberry seed oils from 16 different raspberry varieties and physicochemical properties and, after considering the latest scientific knowledge and market trends, extracted oil from the raspberry seeds of the ‘Polka’ variety using different oil extraction methods. The aim of this study was also to determine the yield and fatty acid content of the oil obtained by different methods and to determine the effect of grounding the seeds into a smaller fraction on oil yield, to compare the yield, colors, fatty acids, kinematic and dynamic viscosity, moisture, and volatile matter content. 

With the rapid development of technology and increasing competition, raspberry growers need to be committed to long-term investment to stay in the market. To increase the value of business entities, which is most affected by capital investments in new technologies, existing and new business areas, and production development, research, and improvement, it is necessary for companies to properly coordinate the capital investment system, plan raw materials and production volumes, and thus select the most suitable investment projects. This study also considered the economic aspects of raspberry production. The results of the analyses will serve as a starting point for expanding and continuing the research on the influence of various factors on raspberry yields and the economic effectiveness of production.

## 2. Results and Discussion

In the food industry, seeds are usually thrown away as waste from the production process, but they can be used to create valuable products using zero-waste processing technologies, thereby reducing losses for producers. When processing raspberries into juice or puree, the pomace of the studied raspberry varieties ranges from 18.9% (‘Toma’) to 7.6% (‘Willamette’) (Figure 1), the average of 16 varieties is 13.1%. The seeds contain quite a lot of oil. According to the literature, the oiliness of seeds of plants of the genus Rubus varies in the range of 12–21%, black currant seeds contain up to 26.15% oil, blackberry seeds contain up to 15.68%, blueberry seeds contain up to 13.27%, and quince seeds contain up to 17.30%. When extracting the seeds of 16 raspberry cultivars grown in Lithuania with the solvent hexane, the average oil yield was 15.50%. The highest oil content was obtained from ‘Volnica’ raspberry seeds (20.40%) and the lowest from ‘Nagrada’ seeds (12.10%). The yield of ‘Polka’ oil, the most widely grown variety in Europe and Lithuania, was 13.50%. Raspberry seed oil yield from different varieties is presented in Figure 2.

Edible seed oil contains saturated (SFA), monounsaturated (MUFA), and polyunsaturated (PUFA) fatty acids and other valuable bioactive substances, such as tocopherols and tocotrienols, polyphenols, phytosterols, lignans, triterpene, carotenoids, and chlorophylls [25]. Polyunsaturated fatty acids are further classified into ω-3, ω-6, and ω-9 fatty acids. Omega-3 and 6 (ω-3 and ω-6) are called essential fatty acids, because they are acquired only through food consumption and cannot be synthesized by mammals on their own. A pilot study on major depressive disorder and cardiovascular disease patients showed a strong association with polyunsaturated fatty acids, and the authors concluded that it confirms its preventive role in these diseases [26]. A total of 16 raspberry varieties (‘Polka’, ‘Zorinka’, ‘Willamette’, ‘Volnica’, ‘Nagrada’, ‘Austrijas remontanta’, ‘Toma’, ‘Helkal’, ‘Novokitajevskaja’, ‘Sputnica’, ‘Canby’, ‘Bristol’, ‘Ariadne’, ‘Malling Seedling’, ‘Peresvet’, and ‘Meeker’) seeds were studied to identify fatty acids. Raspberry seed oil contains 26 fatty acids, of which the most important are linoleic (C18:2) and α-linolenic (C18:3) acids. The amount of linoleic acid (ω-6) in different varieties of raspberry oil ranged from 57.7 (‘Willamette’) to 44.8% (‘Polka’), and the amount of α-linolenic acid (ω-3) ranged from 25.2 (‘Helkal’) to 37.2% (‘Polka’) (Table 1). The third in the amount of raspberry seed oil contains oleic acid (C18:1), and its amount in different varieties of seed oil ranged from 7.8 (‘Bristol’) to 16.9% (‘Helkal’). Linoleic acid is very important for health; it is the most common acid in human epidermis, directly related to the synthesis of ceramides, which support healthy skin barrier function [27,28].

Raspberry seed oil was characterized by a low (close to optimal) ω-6 and ω-3 fatty acid ratio (Figure 3). The optimal ω-6/ω-3 fatty acid ratio is 1:1 to 2:1 [29]; their balance is an important determinant in decreasing the risk for coronary heart disease, arthritis, diabetes, hypertension, cancer, and other autoimmune and possibly neurodegenerative diseases. The ratio of ω-6 to ω-3 fatty acids in raspberry seed oil tested ranged from 1.63:1 (‘Bristol’) to 2.25:1 (‘Willamette’). Of the 16 raspberry cultivars tested, ‘Polka’ seed oil had the least linoleic (ω-6) (44.79%) and the most α-linolenic (ω-3) fatty acids and the best ratio of ω-6 to ω-3 fatty acids—1.2:1. The content of saturated fatty acids in raspberry seed oils amounted to less than 5% of the total fatty acids (Table 1). 

The unique composition of fatty acids and other useful physical and chemical properties indicate the potential for the use of raspberry seed oil in the food, pharmaceutical, cosmetic, and other industries. Raspberry seed oil is rich in ω-3 fatty acids, so it is nutritionally valuable and can be used as a food additive or supplement. It is certain that raspberry seed oil makes a great addition to an organic product because of its fatty acid composition. The ratio of ω-6 (linoleic) to ω-3 fatty acids in the oil is very favorable for nutrition (approximately 1.4:1), while a diet rich in saturated fatty acids increases the risk of obesity, type 2 diabetes, cardiovascular disease, and many other diseases [30,31,32,33]. In addition, the extraction of oil from raspberry seeds would allow the full use of natural resources and thus reduce environmental pollution.

The yield of raspberry seed oil depends on varieties and extraction with different methods and other factors, like how dry the seeds are, how finely they are ground, their temperature, and the pressure during pressing. 

In the second stage, the amount of oil from the raspberry seed variety ‘Polka’ obtained by different methods was determined. The study found that the highest amount of oil was obtained by supercritical extraction (18.81%), and the lowest amount was obtained by cold pressing (11.2%). The separation of extraneous substances and water by centrifugation and filtering from the oil changed the amount of pure oil obtained. The lowest yield of pure oil is obtained by subcritical CO_2_ extraction (7.8%), and the highest yield is obtained using the solvent hexane (13.5%). The results of the study showed that grinding raspberry seeds into a smaller fraction can yield a larger amount of oil. In this case, grinding raspberry seeds through a 1 mm sieve gave an oil yield of 16.79%, while grinding through a 0.5 mm sieve increased the yield to 18.81%, i.e., 2.02% more oil was obtained by supercritical extraction. Crude and pure raspberry seed oil yields extracted with different methods are presented in Table 2. 

Raspberry seed oil is rich in essential fatty acids, primarily linoleic and linolenic acids. Because of its composition, raspberry seed oil possesses superior anti-inflammatory qualities, which makes it a nice addition to face, lip, and sunscreen products [34,35,36]. In practice, it is very important to know the most efficient way and quality of product extraction. In this context, fatty acids in oils extracted in different ways were analyzed. The results revealed that the oil extraction methodology is not significant for the fatty acid content in the oil (Table 3).

Another very important active substance is carotenoids, which cannot be produced by humans, and carotenoid-rich foods and supplements are their main dietary sources. The major carotenoids in foods and the most studied in relation to human health are the three hydrocarbon carotenes: α-carotene, *β*-carotene, and lycopene. *β*-carotene is the most widely distributed and the most important provitamin A carotenoid. 

Raspberry seed oil contains a lot of carotenoids; their total amount, depending on the extraction method, varies from 0.81 mg/100 g (extracted with subcritical CO_2_) to 3.25 mg/100 g (extracted with supercritical CO_2_) (Figure 4). Britton and Khachik (2009) suggested a useful criterion to facilitate the categorization of carotenoid content in a particular food so that the level of a specific carotenoid can be classified into four different concentration groups: low (0–0.1 mg/100 g), moderate (0.1–0.5 mg/100 g), high (0.5–2 mg/100 g), or very high (>2 mg/100 g). 

Regarding the obtained results, it can be said that raspberry seed oils obtained by different extraction methods presented high or very high levels of carotenoid concentration. The best concentration level of carotenoids obtained is from supercritical CO_2_ extraction, and the lowest concentration was detected in the oil obtained by the subcritical CO_2_ method (Figure 4). The lowest amount of *β*-carotene is also found in the oil extracted with subcritical CO_2_ (0.35 mg/100 g), but the percentage of *β*-carotene in the oil extracted in this way is the highest and reaches 43.21% of the total amount of carotenoids.

In addition to carotenoids, another very important group of fat-soluble antioxidants and vitamins are tocopherols. α-tocopherol is also known as vitamin E. Raspberry seed oil contains from 19.8 to 888 mg/100 g of carotenoids depending on the raspberry variety and extraction method [37,38]. Different amounts of tocopherol isomers are found in raspberry variety ‘Polka’ seed oil depending on the oil extraction method (Figure 5). 

Raspberry seed oil contains the most γ-tocopherol, from 16.1 mg/100 g (solvent extraction) to 26.4 mg/100 g (CO_2_ supercritical extraction) (Figure 5). The amount of α-tocopherol is found to be about eight times less than that of γ-tocopherol, from 2.1 mg/100 g (solvent extraction) to 3.2 mg/100 g (CO_2_ supercritical extraction) (Figure 5). The amount of δ-tocopherol is found to be even lower, from 1.1 mg/100 g (solvent extraction) to 1.8 mg/100 g (CO_2_ supercritical extraction). Only traces of β-tocopherol are detected. The extraction method also affects the total amount of tocopherols: the most tocopherols are extracted by CO_2_ supercritical extraction and the least by solvents (hexane), at 31.4 mg/100 g and 19.4 mg/100 g, respectively.

Crude raspberry seed oil is slightly cloudy and yellowish in color. Raspberry seed oil can range from clear yellow to light brown, depending on how the oil is extracted. This yellowish tinge to the oils is given by carotenoids. A yellowish tint is desirable because it gives the oil the characteristic butter-like appearance, especially in the case of oils without the addition of conventional dyes that are often used in the food industry. 

The lightest in tint (L—44.9%) was the oil extracted with subcritical CO_2_. Other extraction methods had little effect on the brightness of the oil. The oil extracted with subcritical CO_2_ was distinguished from the oil extracted by other methods by the lowest *a** color coordinate (*a**—3.5 NBS units), which indicates that its red color component is small and it is neither red nor green. The larger *b** color coordinate (*b**—39.8 NBS units) shows a large dominance of the yellow component compared to the blue one (Figure 5). This is also confirmed by the chroma (C—40.0 NBS units) and the hue angle of 85.0°, which shows that the color of the oil is very close to pure yellow. Meanwhile, the oil extracted by other methods has a greater shade of red, characteristic of the carotenoid lutein.

Refractive index (n) and density (d) are very important physicochemical indicators characterizing oil. There are very few studies characterizing red raspberry seed oil. Parry and co-authors found [38] that red raspberry seed cold-pressed oil had a refractive index of 1.4788 and a density of 0.929 g/mL. It was found that the influence of different extraction methods on the refractive index and density of red raspberry Polka seed oil is not significant (Table 4). The changes in refractive index and oil density during the extraction of raspberry seed oil by different methods probably depend on the fatty acid composition of the extracted oil.

As viscosity changes the flow properties of liquid food and influences the appearance and consistency of a product, this measuring variable is important in most production stages. Viscosity is a major factor in determining the forces that must be overcome when fluids are used in lubrication and transported in pipelines. It controls the liquid flow in such processes as spraying, injection molding, and surface coating. So, this criterion is important for oil producers when choosing equipment for production, packaging, or offering products to customers. In the chemical and cosmetic industry, viscosity testing is a very important parameter for quality control. By measuring the viscosity of products such as toothpaste, cough syrup or ointment, ink, paint, and coatings, manufacturers can predict how products will behave once they are in the hands of the consumer. Analyzing the viscosity to simulate the sample’s processability during production, as well as the application behavior to ensure customer satisfaction, is one of several test methods in quality control. 

Research results show that the dynamic viscosity of raspberry seed oil decreases exponentially (Figure 6, Table 5) with increasing temperature. The coefficient of determination of the temperature dependence of the dynamic viscosity of raspberry seed oil obtained by various methods is high (R^2^ is from 0.988 to 0.991) (Table 5), which indicates a very strong correlation between the dynamic viscosity and temperature. If at a temperature of 0 °C (simulating storage conditions), differences in oil dynamic viscosity are visible depending on the method of oil extraction, then close to human body temperature (which is relevant when using oil as a cosmetic), the dynamic viscosity is practically the same and reaches 22.3 ± 1.45 mPa·s. Analogous regularities are also observed when analyzing kinematic viscosity (Table 5).

For the calculation of variable and fixed costs, average prices in Lithuania were taken: an average salary of 10.49 EUR/h, electricity cost of 0.167 EUR/kWh, water cost of 2.5 EUR/m^3^, packaging cost for oil glass bottle with a label of 0.7 EUR/unit, packaging cost for puree—bag-in-box 3 kg—of 0.9 EUR/pc, and raw material production (raspberries) cost of 3.5 EUR/kg. To calculate other fixed costs of the company’s own production, administration, maintenance, marketing, communication services, and insurance fees were included. When the company chose to buy the service during the production, the following service prices were calculated: seed boning service cost of 0.2 EUR/kg, seed drying service cost of 0.7 EUR/kg, seed milling service cost of 1.5 EUR/kg, oil extraction service with solvent cost of 3.0 EUR/kg, seed cold-pressing cost of 12.0 EUR/kg, seeds CO_2_ extractions service cost of 4.0 EUR/kg, and puree pasteurization and bottling service costs of 1.0 EUR/kg.

The average market price of raspberry seed oil in the EU is 12.5 EUR/100 mL, and that of 100% raspberry puree is 8 EUR/kg. Equipment prices were calculated based on a survey of EU, Chinese, and US standard equipment suppliers and their commercial offers. 

Biorefining 1 ton of raspberry variety ‘Polka’ yielded 85.5% pulp and 14.5% by-product of 9% seeds. The production of puree from one ton of raspberries leaves an average of 90 kg of seeds. After drying (till the moisture content of seeds is 8.64%), about 34% of the material remains, which is 30.6 kg. 

Different production methods have revealed the need to assess the potential risks of investing in raspberry biorefining processes (oil and puree production). The maximum oil content of 4131 mL is obtained by using the solvent hexane. It has been found that investing in equipment requires a minimum of 29.04 kg raspberry seeds to be processed to pay off the annual depreciation of the equipment, but such oil is negatively evaluated by scientists and consumers due to the possible residues of the chemical solvent in the oil, which may cause the company to have difficulties in marketing such a product. 

The lowest oil content is obtained by extracting raspberry seeds by subcritical CO_2_. This method also stands out as one of the highest depreciations of fixed assets. A total of 119,920 mL of oil is required to cover the annual depreciation costs of the equipment.

The most expensive method of oil production is supercritical CO_2_ extraction. As much as 159,920 mL of oil is needed to cover the annual cost of this equipment. The most optimal way to extract the oil is by cold pressing, because it is obtained in the “green” way, the payback costs of the equipment are not high, it takes the least time to produce it, and no special knowledge is required during operation.

From the data obtained, it can be concluded that it would be unprofitable for a company to invest in fixed assets to produce a small amount of production per year, as the profit obtained does not fully cover the depreciation costs. Profits can be made by choosing to purchase a service, or the company should provide the services itself and produce larger quantities to offset the costs. 

The yield of oil and puree from the raspberry variety ‘Polka’, the costs of producing products in economically different ways, the cost of equipment and annual depreciation, the cost of production of oil and puree services, and the potential sales revenue, profit, and the need for raw material to cover the annual depreciation costs of the equipment are given in Table 6.

## 3. Materials and Methods

### 3.1. Plant Material and Its Preparation 

In the first stage, raspberry seeds (*Rubus idaeus* L.) were of the 16 varieties: ‘Polka’, ‘Austrijas Remontanta’, ‘Bristol’, ‘Volnica’, ‘Willamette’, ‘Malling Seedling’, ‘Ariadne’, ‘Novokitajevskaja’, ‘Meeker’, ‘Helkal’, ‘Zorinka’, ‘Toma’, ‘Peresvet’, ‘Sputnica’, ‘Nagrada’, and ‘Canby’. After pressing the berries, the pomace was collected and dried in a convection dryer (thickness of approx. 0.5 cm) at temperature (40 °C) for 24 h, with occasional stirring. Raspberry seeds were ground in an ultra-centrifugal rotor mill ZM200 (Retsch, Haan, Germany) using 0.2 mm particle size sieve, but the process was stopped for 15 s at 15–30 s intervals to avoid heating the sample. 

In the second stage, the Polish primocane raspberry variety ‘Polka’ was selected for more detailed research. This variety has been chosen as currently the most popular and one of the main cultivated raspberry varieties grown in the world, with excellent quality dessert berries and a rich harvest. This variety has also attracted a great deal of interest from scientists. 

Fresh raspberries were taken from Audrones Ispiryan’s Lithuanian national quality certified farm (GPS coordinates: 55°47′42.2″ N 22°44′59.0″ E). The cultivation process on the farm is distinguished by its naturalness, nutrition, and environmental aspects. It has limited use of protective equipment and no use of environmentally harmful plant protection products. Raspberry seeds were obtained by separating them by using de-stoning machine EP500 (VORAN Maschinen GmbH, Pichl, Austria). The seeds were dried naturally at approximately 25–28 °C and grounded in a rotary beater mill SR 300 (Retsch, Germany) using a 1 mm sieve (with an average particle size of 1 mm) and stored in hermetically closed glass jars in a dark, dry room until the oil was extracted. To determine the effect of raspberry seed grounding on oil yield, seeds were grounded using 1 mm, 0.75 mm, and 0.5 mm sieves (with an average particle size of 1 mm, 0.75 mm, and 0.5 mm). The moisture content of seeds was 8.64%. Humidity was determined with moisture analyzer MOC63u (Shimadzu).

In the third stage, oil was extracted from raspberry seeds via 4 different methods: solvent extraction, cold extraction/pressing, extraction with subcritical CO_2_, and extraction with supercritical CO_2_. 

### 3.2. Solvent Extraction

A total of 1 kg of the ground raspberry seeds were placed in a 3 L glass vessel and filled with hexane. The extraction is carried out for 24 h at a temperature of 25 ± 2 °C in dark with stirring. The solvent was removed by vacuum filtration, and the sample was extracted twice. After the last filtration, the extract was pooled, hexane was removed with a vacuum rotary evaporator Rotavapor R-205 (BÜCHI Labortechnik AG, Flawil, Switzerland) at 35 ± 2 °C and 170 mbar pressure, purged with nitrogen, and stored at −18 °C until analysis. 

### 3.3. Cold Extraction/Pressing 

Cold extraction/pressing of the oil was carried out with the cold pressing Machine PR-H100/1 (1Head) (Oil press GmbH & Co. KG, Reut, Germany) at a speed of 10 Hz (8 RPM), capacity of 2.38 kg/h, and yield of oil of 0.3 kg/h. The oil was extracted from 9 kg of raspberry seeds and stored at −18 °C until analysis.

### 3.4. Extraction with Subcritical CO_2_

The oil was extracted with subcritical CO_2_ at a pressure of 5 MPa and a temperature of 10 °C for 16 h with a subcritical extractor Eco Extractum, Lithuania [39]. Pure oil was purged with nitrogen and stored at −18 °C until analysis.

### 3.5. Extraction with Supercritical CO_2_

The supercritical extraction experiments were carried out using supercritical fluid extractor SFT-150 (Supercritical Fluid Technologies, Newark, DE, USA). Each extraction was performed using 50 g of ground dried raspberry seed sample. Each sample was loaded into 150 mL thick-walled stainless cylindrical extractor vessel with 5-micron frits. The temperature (60 ± 2 °C) of the extraction vessel was controlled by a surrounding heating jacket. The volume of CO_2_ consumed was measured by a gas flow meter Gallus 2000 (Schlumberger Industries, Guebwiller, France) and expressed in standard liters per minute (SL/min). Flow rates were 1.4 SL/min. The process consisted of static (120 min) and dynamic (300 min) extraction steps. The static extraction time was included in the total extraction time of 420 min.

Pure oil was purged with nitrogen and stored at −18 °C until analysis.

### 3.6. Chemicals, Solvents, and Gasses

CO_2_ and N_2_ were obtained from Gaschema (Jonava, Lithuania); α-tocopherol, β-tocopherol, γ-tocopherol, δ-tocopherol, β-carotene, and hexane were obtained from Sigma-Aldrich (Steinheim, Germany).

### 3.7. Determination of Fatty Acid

Fatty acid composition for raspberry seed oil was determined as described by ISO 12966-1: 2015 [40]; ISO 12966-2: 2017 [41]. All oil analyses were performed with filtered oil. 

### 3.8. Determination of Carotenoids

The total amount of carotenoids and *β*-carotene content were determined by HPLC method according to [17], with slight modifications. The oil samples (1.0 ± 0.01 g) were extracted with a 10 mL of n-hexane containing 1% butylhydroxytoluene (BHT), then filtered through a 0.45 mm polyvinylidene fluoride (PVDF) syringe filter (Millipore, Burlington, MA, USA). The total carotenoids and *β*-carotene contents were analyzed using the HPLC method on a Waters HPLC system consisting of 2695 liquid separation module, UV–Vis detector UV–Vis 2489 (Waters Corporation, Milford, MA, USA), and equipped with an RP-C30 column, (5 μm, 4.6 × 250 mm, YMC™ Europe, Dinslaken, Germany) connected to a C30 guard column (5 μm, 10 × 4.0 mm, YMC Europe, Dinslaken, Germany). The flow rate was 0.65 mL/min, column temperature was 22 °C, and *β*-carotene was detected at 450 nm. The mobile phase consisted of methanol (solvent A) and methyl-tert-butyl ether (solvent B). The samples were injected at 1% B (held 1 min), and the gradient then changed to 100% B (1−90 min) and again to 1% B in 5 min (held 5 min). For quantification, a calibration curve was produced using an authentic all-trans-*β*-carotene standard (concentration range was from 0.1 to 5.0 mg/100 mL).

### 3.9. Determination of Tocopherols

Analysis of tocopherols was performed by HPLC according to Brazaityte et al.’s [25] methodology with some modifications. About 1 mg of oil was weighed in an Eppendorf tube, and then 1 mL of n-hexane with 1% of BHT was added to the tube. Afterward, the samples were filtrated through a 0.45 µm polytetrafluoroethylene (PTFE) membrane syringe filter (VWR International, Radnor, PA, USA) and were analyzed by HPLC/FLD (fluorescence detector) (Agilent Technologies, Santa Clara, CA, USA). The HPLC measurements were performed using a normal phase column (Phenomenex Luna Silica, 5 μm, 250 mm × 4.6 mm). The HPLC 10A system, equipped with an RF-10A fluorescence detector (Shimadzu, Japan), was used for analysis. Peaks were detected at an excitation wavelength of 295 nm and an emission wavelength of 330 nm. The mobile phase (0.5% isopropanol in hexane) was used at a flow rate of 1 mL min^−1^. The α-tocopherol, γ-tocopherol, and δ-tocopherol were identified according to the analytical standard. The α-tocopherol, γ-tocopherol, and δ-tocopherol content were expressed per 100 g of oil.

### 3.10. Color Measurement

The color coordinates of the oil samples in the CIE L*a*b* color space were measured with a MiniScan XE Plus spectrophotometer (Hunter Associates Laboratory, Inc., Reston, VA, USA). The parameters evaluated during reflected-color measurements were L*, a*, and b* (brightness and red and yellow coordinates according to the CIE L*a*b* scale, respectively), and color saturation (the chroma value) was calculated (C = (a*^2^ + b*^2^)^1/2^). The values L*, a*, b*, and C* were measured in NBS units. The NBS unit is a unit of the U.S. National Bureau of Standards and meets one color resolution threshold, i.e., the smallest difference in a color that can be captured by a trained human eye. Prior to each series of measurements, the spectrophotometer was calibrated with a light trap and a white standard with the following color coordinates in the XYZ color space: X = 81.3, Y = 86.2, and Z = 92.7. The value of L* indicated the ratio of white to black, the value of a* indicated the ratio of red to green, and the value of b* indicated the ratio of yellow to blue. Five replications were taken for the analysis. The color coordinates were processed by the Universal Software V. 4-10.

### 3.11. Determination of Kinematic and Dynamic Viscosity 

The dynamic viscosity of raspberry oil was determined with a Höeppler Viscometer’ B3 (Carl Zeiss Jena, Leipzig, Germany) using a Hoeppler falling ball viscometer in a 10° cylinder filled with oil. The dynamic viscosity is calculated according to the equation
η = t · (ϱ_1_ − ϱ_2_) · K,
where

η—viscosity, mPa·s;

t—time of descent from top to bottom annular mark, s;

ϱ_2_—density of liquid, g/cm^3^;

ϱ_1_—density of ball, g/cm^3^;

K—ball constant, mPa·cm^3^/g.

Thermostating was performed with refrigerating/heating circulators PolyScience Model 912 (Niles, IL, USA), at temperature stability ±0.1 °C.

### 3.12. Assay of Raspberry Oil Density 

Density was determined picnometrically according to AOAC Official Method 9201.212 at 20 °C. The accuracy of the density determination was about 1 × 10^–4^ g/cm^3^.

### 3.13. Determination of Refractive Index

The refractive index was determined according to ISO 6320:2017 at 20 ± 0.1 °C temperature with a Carl Zeiss Abbé refractometer Model I. The measurement was repeated five times.

### 3.14. Determination of Need for Raw Material to Cover the Annual Depreciation Costs of the Equipment 

A cost–benefit analysis method was used to assess the economic efficiency of the technologies. A cost–benefit analysis (CBA) is the process used to measure the benefits of a decision or taking action minus the costs associated with taking that action. A cost–benefit analysis is a systematic process that businesses use to analyze which decisions to make and which to forgo. The cost–benefit analysis sums up the potential rewards expected from a situation or action and then subtracts the total costs associated with taking that action. CBA involves measurable financial metrics—as revenue earned or costs saved as a result of the decision to pursue the project of a raspberry (*Rubus idaeus* L.) seed oil extracted in various ways. The economic situation was modeled, and the costs were calculated on the example of Lithuanian raspberry farms and processing companies. The potential sales revenue is calculated by taking the average price of oil in Europe per 100 mL/EUR 12.5 and the price of 100% puree, 1 kg/EUR 8. Calculations were performed by biorefining (dividing one product into several separate) raspberries into raspberry pulp (product—100% raspberry puree) and seeds (product—oil). The costs of two economically different production processes were compared. Depreciation of equipment was calculated using the straight-line method of depreciation and is calculated using the following formula:N = (V_1_ − V_2_)/T
where

N—annual depreciation amount;

V_1_—acquisition value of tangible fixed assets (cost of production);

V_2_—the liquidation value of tangible fixed assets;

T—useful life of equipment in years.

### 3.15. Statistical Analysis 

For the statistical processing of the data obtained from the analysis of the chemical composition of the oil, means and standard deviations were calculated with STATISTICA 10 StatSoft, Inc., Tulsa, OK, USA) and Excel (Microsoft, Redmond, WA, USA) software. One-way analysis of variance (ANOVA) along with post hoc Tukey’s HSD test was employed for statistical analysis. Differences were considered to be significant at *p* < 0.05.

## 4. Conclusions

The results demonstrate the potential of raspberry by-products focusing on mainstream sectors such as the food, nutraceutical, pharmaceutical, and cosmetic industries. Raspberries bioactive compounds are known to be beneficial for health and can be utilized in cosmetic or pharmaceutical industries.

All studies have been conducted with unfiltered oil to see if there is a misleading view that oils extracted in one way or another may be more valuable, of better quality, etc. In this case, the fatty acid composition does not change significantly. It can also be concluded that in research on oil filtration, the determination of the quantity and quality of pure oil would be relevant.

This research has a very important practical significance as it has revealed that processors wishing to place raspberry seed oil on the market because of its omega acid properties should opt for a cold oil production method, as the investment in equipment is significantly lower and the processing time is lower, resulting in the lowest cost and quality of the product using the market ways of extracting oil.

It can be concluded that in order to maximize profits and the value of their property, companies must invest and implement only those projects whose average expected return is higher than the cost of capital. With the data obtained during the study, raspberry processors can prepare an investment project by calculating whether the equipment in which the investment will be used will be used efficiently. The economic evaluation revealed the potential benefits of the project in relation to the cost of capital, and the company, having the data identified in the study, can model them with its own data to estimate the potential cost of breaking even, which would show how much production is needed to bring sales revenue into line with their production costs (variable and fixed). These measures can be used to decide which investment projects to finance and what priority to give them to ensure that the optimal return on investment will be obtained and the risk of the project will be minimal.

## Figures and Tables

**Figure 1 plants-12-02706-f001:**
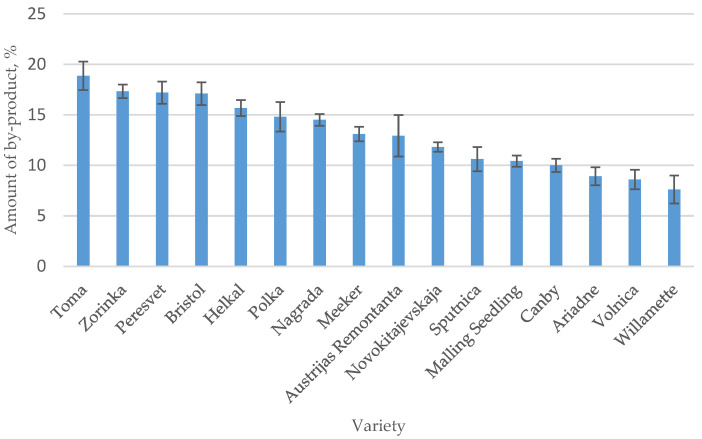
Amount of by-products of different raspberry berry varieties.

**Figure 2 plants-12-02706-f002:**
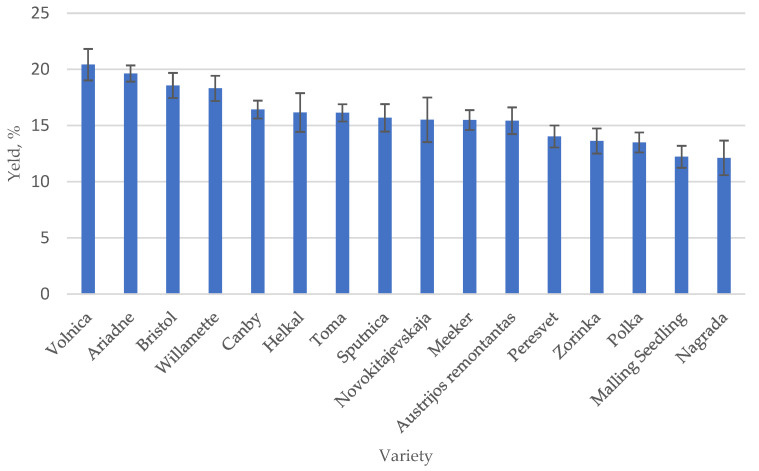
Oil yields from the raspberry seeds.

**Figure 3 plants-12-02706-f003:**
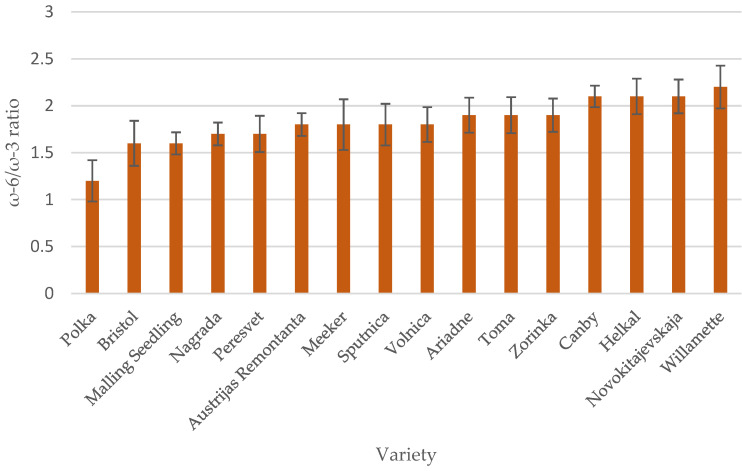
ω-6/ω-3 fatty acid ratio of raspberry seed oil of different varieties.

**Figure 4 plants-12-02706-f004:**
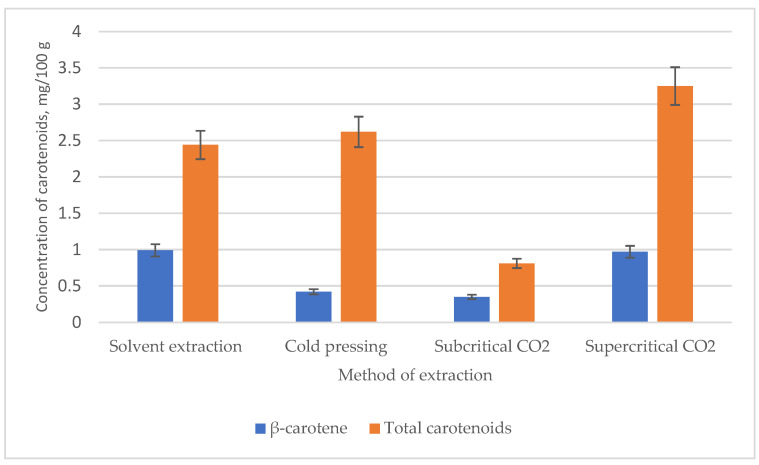
Effect of different extraction methods on the total carotenoids and *β*-carotene content.

**Figure 5 plants-12-02706-f005:**
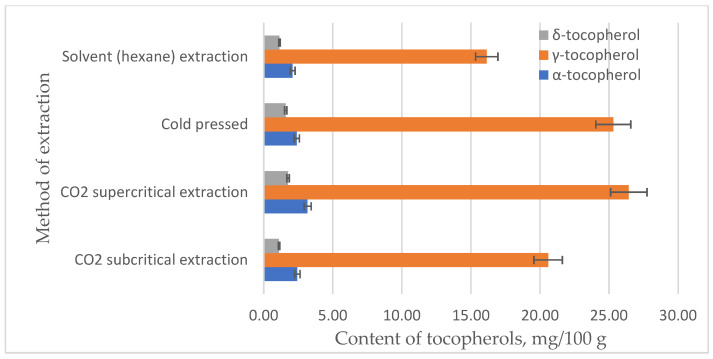
Effect of different extraction methods on the tocopherol isomers content in raspberry seed oil.

**Figure 6 plants-12-02706-f006:**
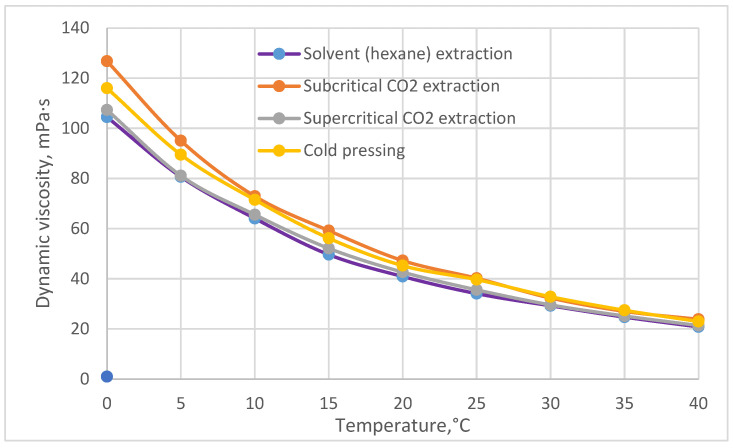
Dynamic viscosity of raspberry seeds oil extracted by various methods.

**Table 1 plants-12-02706-t001:** Fatty acid composition of raspberry seed oil of different varieties, %.

Raspberry Varieties	α-Linolenic Acid (ω-3)	Linoleic Acid (ω-6)	Oleic Acid (ω-9)	Palmitic Acid
‘Ariadne’	28.8 ± 2.1 ^bcd^	54.1 ± 2.5 ^a^	11.8 ± 0.8 ^b^	2.6 ± 0.2 ^ab^
‘Austrijas Remontanta’	29.9 ± 2.0 ^bcd^	53.8 ± 2.2 ^a^	10.4 ± 1.0 ^bc^	2.8 ± 0.3 ^ab^
‘Bristol’	33.2 ± 2.0 ^ab^	54.3 ± 2.1 ^a^	7.8 ± 0.9 ^c^	1.9 ± 0.2 ^ab^
‘Canby’	26.5 ± 1.7 ^d^	55.8 ± 2.4 ^a^	11.1 ± 0.9 ^bc^	3.0 ± 0.2 ^a^
‘Helkal’	25.2 ± 2.0 ^d^	53.0 ± 2.9 ^a^	16.9 ± 1.2 ^a^	2.1 ± 0.2 ^ab^
‘Malling Seedling’	32.5 ± 2.1 ^abc^	51.4 ± 2.9 ^ab^	10.3 ± 1.9 ^bc^	2.5 ± 0.5 ^ab^
‘Meeker’	29.7 ± 1.9 ^bcd^	54.4 ± 2.0 ^a^	10.2 ± 1.0 ^bc^	2.8 ± 0.6 ^ab^
‘Nagrada’	30.5 ± 2.2 ^bcd^	51.4 ± 2.2 ^ab^	12.3 ± 0.9 ^b^	2.5 ± 0.4 ^ab^
‘Novokitajevskaja’	27.0 ± 1.8 ^cd^	56.9 ± 1.6 ^a^	10.5 ± 1.3 ^bc^	2.6 ± 0.4 ^ab^
‘Peresvet’	30.8 ± 2.1 ^bcd^	53.8 ± 2.2 ^a^	9.9 ± 1.0 ^bc^	2.5 ± 0.4 ^ab^
‘Polka’	37.2 ± 1.9 ^a^	44.8 ± 2.0 ^b^	10.6 ± 1.0 ^bc^	1.8 ± 0.3 ^b^
‘Sputnica’	30.3 ± 2.1 ^bcd^	54.9 ± 2.1 ^a^	9.7 ± 1.1 ^bc^	2.4 ± 0.4 ^ab^
‘Toma’	29.5 ± 1.9 ^bcd^	55.4 ± 2.1 ^a^	10.1 ± 1.0 ^bc^	2.4 ± 0.4 ^ab^
‘Volnica’	30.1 ± 1.9 ^bcd^	54.3 ± 2.3 ^a^	10.0 ± 1.1 ^bc^	2.6 ± 0.6 ^ab^
‘Willamette’	25.7 ± 2.1 ^d^	57.7 ± 2.1 ^a^	11.2 ± 1.1 ^b^	2.5 ± 0.5 ^ab^
‘Zorinka’	28.0 ± 1.0 ^bcd^	54.1 ± 2.0 ^a^	12.0 ± 2.0 ^b^	2.6 ± 0.4 ^ab^

Values are expressed as the mean ± standard deviation (*n* = 3). Values with different letter superscripts at different columns are considered significantly different at *p* < 0.05.

**Table 2 plants-12-02706-t002:** Oil yield of raspberry variety ‘Polka’ seeds extracted with different methods.

Method of Extraction	Crude Oil Yield, %	Pure Oil Yield, %
Solvent (hexane)	13.5 ± 0.4 ^a^	13.5 ± 0.7 ^d^
Cold pressing	11.2 ± 0.3 ^b^	9.1 ± 0.3 ^e^
Subcritical CO_2_	11.9 ± 0.5 ^c^	7.8 ± 0.4 ^a^
Supercritical CO_2_ (particle size of 1 mm)	16.79 ± 0.6 ^d^	12.1 ± 0.4 ^b^
Supercritical CO_2_ (particle size of 0.75 mm)	17.98 ± 0.6 ^e^	12.6 ± 0.6 ^b^
Supercritical CO_2_ (particle size of 0.5 mm)	18.81 ± 0.8 ^f^	12.9 ± 0.5 ^bc^

Values are expressed as the mean ± standard deviation. Values with different letter superscripts at different columns are considered significantly different at *p* < 0.05.

**Table 3 plants-12-02706-t003:** Fatty acid composition of seed oil (variety ‘Polka’) extracted with different methods, g/100 g.

Extraction Method	Linoleic (ω-6)	α-Linolenic (ω-3)	Oleic (ω-9)	Palmitic
Solvent (hexane) extraction	44.8 ± 2.0 ^a^	37.2 ± 1.9 ^a^	10.6 ± 1.0 ^a^	1.8 ± 0.3 ^a^
Cold pressing	44.8 ± 0.9 ^a^	37.7 ± 0.6 ^a^	10.4 ± 0.7 ^a^	1.9 ± 0.2 ^a^
Supercritical CO_2_ extraction	44.0 ± 0.7 ^a^	37.9 ± 0.8 ^a^	10.2 ± 0.7 ^a^	2.1 ± 0.3 ^a^
Subcritical CO_2_ extraction	44.0 ± 0.4 ^a^	38.1 ± 1.1 ^a^	10.3 ± 0.9 ^a^	2.1 ± 0.6 ^a^

Values are expressed as the mean ± standard deviation. Values with different letter superscripts at different columns are considered significantly different at *p* < 0.05.

**Table 4 plants-12-02706-t004:** Effect of extraction method on the refractive index and density of red raspberry variety ‘Polka’ seed oil.

Method of Extraction	Refractive Index at 20 °C	Oil Density, g/cm^3^
Solvent (hexane) extraction	1.4832 ± 0.22 ^a^	0.9258 ± 0.19 ^a^
Cold pressing	1.4831 ± 0.23 ^a^	0.9311 ± 0.21 ^a^
Supercritical CO_2_ extraction	1.4829 ± 0.19 ^a^	0.9334 ± 0.21 ^a^
Subcritical CO_2_ extraction	1.4835 ± 0.17 ^a^	0.9366 ± 0.18 ^a^

Values are expressed as the mean ± standard deviation. Values with different letter superscripts in different columns are considered significantly different at *p* < 0.05.

**Table 5 plants-12-02706-t005:** Temperature dependence of dynamic and kinematic viscosity of Polka raspberry seed oil extracted by various methods.

Method of Extraction	Dynamic Viscosity, mPa·s		Kinematic Viscosity, mm^2^·s^−1^	
	Regression Equation	Coefficient of Determination R^2^	Regression Equation	Coefficient of Determination R^2^
Solvent (hexane) extraction	y = 96.527e^−0.04x^	0.991	y = 104.98e^−0.04x^	0.991
Cold pressing	y = 107.91e^−0.04x^	0.991	y = 116.2e^−0.04x^	0.991
CO_2_ supercritical extraction	y = 99.178e^−0.04x^	0.990	y = 106.79e^−0.04x^	0.991
CO_2_ subcritical extraction	y = 115.49e^−0.042x^	0.988	y = 128.46e^−0.043x^	0.987

**Table 6 plants-12-02706-t006:** Economic indicators of raspberry processing.

Raspberry Production	RSO Solvent (Hexane) Oil Extraction	RSO Cold Pressing of Oil	RSO CO_2_ Subcritical Oil Extraction	RSO CO_2_ Supercritical Oil Extraction	Puree Production
Production yield from 1 ton of raspberries (30.6 kg dry molded seeds and 855 kg puree) mL	*Pure oil* *4131*	*Pure oil* *3397*	*Pure oil* *2387*	*Pure oil* *3703*	*Puree* *(100% berries)* *855,000*
**Independent production from 1 ton of raspberries**					
Investment in equipment	*Solvent extraction equipment (2000 mL) is EUR 5000*	*Oil press machine (10–30 kg seeds per hour) is EUR 13,000*	*10 L extractor is EUR 150,000*	*10 L extractor is EUR 200,000*	*Heating 150 L tank with volumetric filling machine* *EUR 45,000*
*1.* **Fixed costs per year**					
1.1. Annual depreciation amount of equipment	*490*	*1290*	*14,990*	*19,990*	*4490*
1.2. Other fixed costs per year	*15,000*	*15,000*	*15,000*	*15,000*	*15,000*
*2.* **Variable costs**					
2.1. Working hours 2.2. salary	*9* *94.41*	*2* *41.96*	*16* *167.84*	*7* *73.43*	*8* *83.92*
2.3. Electricity and water	*32.56*	*15.86*	*55.94*	*25.88*	*29.22*
2.4. Packaging	*28.70*	*23.80*	*16.80*	*25.90*	*270.00*
2.5. Raw material	*350.00*	*350.00*	*350.00*	*350.00*	*3150.00*
**Total**	** *15,995.67* **	** *16,721.62* **	** *30,580.58* **	** *35,465.21* **	** *23,023.14* **
**Production of products by purchasing a service of 1 ton of raspberries**					
Seed boning service	*180*	*180*	*180*	*180*	*180*
Seed drying, milling, and extraction service	*201*	*170*	*231*	*231*	*-*
Puree pasteurization and bottling service	*-*	*-*	*-*	*-*	*900*
Packaging	*29*	*24*	*17*	*26*	*270*
Raw material	*350*	*350*	*350*	*350*	*3150*
**Total**	** *759* **	** *724* **	** *778* **	** *787* **	** *4500* **
**Summary and comparison of economically different productions**					
**Potential sales revenue (average market price in EU)**	*516*	*425*	*298*	*462*	*7200*
**Profit (loss)** **(Independent production)**	*−15,479*	*−16,296*	*−30,282*	*−35,003*	*−15,823*
**Profit (loss)** **(service)**	*−243*	*−300*	*−480*	*−325*	*2700*
**The amount of production** required to cover the depreciation costs of the equipment in mL	*3920*	*10,320*	*119,920*	*159,920*	*62,361*
**The need for raw material** to cover the annual depreciation costs of the equipment (dry molded seeds/kg)	*29*	*93*	*1537*	*1322*	*62,361*

## Data Availability

The data presented in this study are available on request from the corresponding author.

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
