# Peer review of "Physico-Chemical Properties, Fatty Acids Profile, and Economic Properties of Raspberry (Rubus idaeus L.) Seed Oil, Extracted in Various Ways"

_plants, 2023, doi:10.3390/plants12142706_

Round 1

Reviewer 1 Report

Overview and general recommendation:

Thanks for the opportunity to review this research. The manuscript entitled Physico-chemical properties, fatty acids profile and economic properties of raspberry (Rubus idaeus L.) seed oil, extracted in various ways” demonstrates the potential of raspberries seeds oils from 16 different raspberry varieties and physicochemical properties and considering the latest scientific knowledge and market trends, oil was extracted from the raspberry seeds of the ‘Polka’ variety using different oil extraction methods. The subject of the manuscript is topical and the article is very interesting to read. The chosen approaches for extracting raspberry oil are innovative for producers.

1.         Introduction

The introduction is well structured, but the authors should also highlight the importance of the study.

2.         Materials and Methods

The used methods are accurate.

4.         Results and discussion

The results are clearly presented. All characters in the text must be in italic.

There are some technical errors in the text and the references. Please check them. The length, quality and language of the paper are adequate. I recommend the publishing of the paper after the necessary minor corrections.

Author Response

Thank You for such good and beneficial comments. The all suggestions were included in the article. We’ve:

  1. highlighted the importance of the study in the introduction.
  2. Changed characters in italic.
  3. Revised the references and did some changes in all the paper.

Reviewer 2 Report

This work reports on physico-chemical properties, fatty acids profile and economic properties of raspberry seed oil, extracted in various ways. The article is generally informative and well written but there are some points that must be improved before publication.

The English language should be refined, many sentences are imprecisely expressed. Also, there are many grammatical errors throughout the proof. Therefore, this entire manuscript should be carefully checked.

The efficiency of supercritical extraction depends on the process conditions. The content of extracted tocopherols and carotenoids also depends on the conditions (temperature, pressure, time and others). Where do these parameters come from (p. 2.5). Has optimization been done?

There is no methodology for the determination of tocopherols.

A few minor remarks:

Line 87: [18] shouldn't it be at the end of the sentence?

Line 153: this is “recovery” or yield?

Line 186-187: what are these numbers for? 2695 and 2489

Line 230-231: do not repeat the number of the standard in brackets or add it to the references

In Table 1 and other: the number of decimal places must be the same in the result and in standard deviation

Line 380: what does the oil contain?

Author Response

Thank You for the constructive and beneficial comments. The all suggestions were included in the article. Please, find our answers to your comments in the table below.

The English language should be refined, many sentences are imprecisely expressed. Also, there are many grammatical errors throughout the proof. Therefore, this entire manuscript should be carefully checked.

The revision in all aspects was done in the article, with the consultation of native English speakers and all co-authors.

The efficiency of supercritical extraction depends on the process conditions. The content of extracted tocopherols and carotenoids also depends on the conditions (temperature, pressure, time and others). Where do these parameters come from (p. 2.5). Has optimization been done? There is no methodology for the determination of tocopherols.

The standard conditions were used for seeds separation. In the literature the standard conditions for seeds extraction with supercritical CO2 is: temperature 40-60 °C; pressure 250-350 bar; time from 150 to 420 min. The main idea was to compare the oil from raspberry seeds extracted by 4 different methods: solvent extraction, cold ex-traction/pressing, extraction with subcritical CO2, extraction with supercritical CO2.

We added the methodology in the article.

Line 87: [18] shouldn't it be at the end of the sentence?

We’ve corrected it.

Line 153: this is “recovery” or yield?

Yield. We’ve corrected it

Line 186-187: what are these numbers for? 2695 and 2489

Waters HPLC system detectors module numbers, UV–vis detector UV–vis2489 and diode-array detector (DAD, DAD-2998 (Water Corporation (Waters Corporation, Milford).

Line 230-231: do not repeat the number of the standard in brackets or add it to the references

We’ve corrected it

In Table 1 and other: the number of decimal places must be the same in the result and in standard deviation

We’ve corrected it

Line 380: what does the oil contain?

Carotenoids. We’ve corrected.

Reviewer 3 Report

The article entitled "Physico-chemical Properties, Fatty Acids Profile and Economic  Properties of Raspberry (Rubus idaeus L.) Seed Oil, Extracted in Various Ways " presents a quite serious and rigorous study. The authors have done a great job. The authors have studied the optimal variables for extracting raspberry seed oil. The results show that it is possible to extract the bioactive compounds present in raspberries and that they can be used in the cosmetic and pharmaceutical industries. These results are very important for producers wishing to commercialize raspberry seed oil. However, the authors should clarify some points of the paper as follows:

1.- Line 119.- Is all the dried pomace raspberry seed? How have the seeds been separated from other possible parts of the raspberry?

2.- Line 121.- How has this extraction been carried out?

3.- Line 123.- Here it is indicated that we are going to work only with the Polka variety. However, results are then predicted for all varieties. I do not understand.

4.- Lines 132-136.- Why is this oil extraction procedure now indicated? Between lines 117 and 122 another procedure is indicated.

Lines 138-139.- How was the humidity determined?

5.- Section 2.2.- How much seed was added to the vessel? What amount of hexane was added? What was the volume of the container? What agitation system was used?

6.- Section 2.3.- Explain better. Give more technical details. How much seed? How was the extract stored? ......

7.- Lines 169-171.- I do not understand. Why are the samples stored first at 0ºC and then at -18ºC? When they are extracted with hexane they are stored at -18ºC.

8.- Line 196.- How have the LOD and LOQ been calculated. Why have they not been calculated for fatty acids?

 For all these reasons, I consider that the article needs a minor revision.

Author Response

Thank You for the constructive and beneficial comments. The all suggestions were included in the article. Please, find our answers to your comments below.

Line 119.- Is all the dried pomace raspberry seed? How have the seeds been separated from other possible parts of the raspberry?

Yes. Raspberries are usually sorted in production before they are pitted, so there are no other possible parts. Raspberries are also the only fruit that is not washed before production.

Line 121.- How has this extraction been carried out?

The oil was extracted from raspberry seeds by 4 different methods: solvent extraction, cold ex-traction/pressing, extraction with subcritical CO2, extraction with supercritical CO2.

The detailed methods is explained in 2.2, 2.3, 2.4 and 2.5 parts.

Line 123.- Here it is indicated that we are going to work only with the Polka variety. However, results are then predicted for all varieties. I do not understand.

We’ve indicated that in the second stage we will work only with ‘Polka’. With all varieties we’ve worked in the first stage.

Lines 132-136.- Why is this oil extraction procedure now indicated? Between lines 117 and 122 another procedure is indicated.

Because of different stages.

Lines 138-139.- How was the humidity determined?

We’ve added it in the line 140-141

Section 2.2.- How much seed was added to the vessel? What amount of hexane was added? What was the volume of the container? What agitation system was used?

We’ve added data to the section 2.2.

Section 2.3.- Explain better. Give more technical details. How much seed? How was the extract stored?

We’ve added data to the section 2.3

Lines 169-171.- I do not understand. Why are the samples stored first at 0ºC and then at -18ºC? When they are extracted with hexane they are stored at -18ºC.

As there were sediments in the extracted oil, we had to eliminate them. For this, we cooled the oil to 0 °C. After separating the precipitate, the oil was frozen to 18 °C until further analyses. During oil extraction with hexane, the precipitate could not be removed by centrifugation, it was evaporated.